# Exploring the Structural Rearrangements of the Human Insulin-Degrading Enzyme through Molecular Dynamics Simulations

**DOI:** 10.3390/ijms23031746

**Published:** 2022-02-03

**Authors:** Mariem Ghoula, Nathalie Janel, Anne-Claude Camproux, Gautier Moroy

**Affiliations:** 1Unité de Biologie Fonctionnelle et Adaptative, CNRS, INSERM, Université de Paris, F-75013 Paris, France; mariem.ghoula@inserm.fr; 2Unité de Biologie Fonctionnelle et Adaptative, CNRS, Université de Paris, F-75013 Paris, France; nathalie.janel@u-paris.fr

**Keywords:** molecular dynamics simulation, insulin-degrading enzyme, therapeutic target

## Abstract

Insulin-degrading enzyme (IDE) is a ubiquitously expressed metallopeptidase that degrades insulin and a large panel of amyloidogenic peptides. IDE is thought to be a potential therapeutic target for type-2 diabetes and neurodegenerative diseases, such as Alzheimer’s disease. IDE catalytic chamber, known as a crypt, is formed, so that peptides can be enclosed and degraded. However, the molecular mechanism of the IDE function and peptide recognition, as well as its conformation changes, remains elusive. Our study elucidates IDE structural changes and explains how IDE conformational dynamics is important to modulate the catalytic cycle of IDE. In this aim, a free-substrate IDE crystallographic structure (PDB ID: 2JG4) was used to model a complete structure of IDE. IDE stability and flexibility were studied through molecular dynamics (MD) simulations to witness IDE conformational dynamics switching from a closed to an open state. The description of IDE structural changes was achieved by analysis of the cavity and its expansion over time. Moreover, the quasi-harmonic analysis of the hinge connecting IDE domains and the angles formed over the simulations gave more insights into IDE shifts. Overall, our results could guide toward the use of different approaches to study IDE with different substrates and inhibitors, while taking into account the conformational states resolved in our study.

## 1. Introduction

Insulin-degrading enzyme (IDE), also known as Insulysin, is a zinc protease of the M16 metalloprotease family [1,2,3,4,5]. IDE was originally discovered and named since it is the major enzyme responsible for insulin degradation, in vitro, and insulin binding with high affinity (~10 nM) [6]. IDE plays a major role in preventing type II diabetes [6,7,8,9] and other diseases, such as Alzheimer’s [10,11,12,13,14,15,16]. These characteristics make it one of the most important enzymes in the human body. Moreover, IDE directly links with these diseases, making it a promising therapeutic target to design efficient regulators [17,18]. IDE rapidly breaks down insulin and other peptides to prevent toxic amyloid formation [2,4,6,10]. An important feature of IDE is its large cavity (~15,000 A^3^), where peptides are degraded based on their size, charge distribution, and amyloidogenic nature [19,20,21,22,23,24]. IDE is a catalytic protein known to switch between a closed and an open state. The transition from the closed to the open conformation is triggered by the achievement of a specific interaction between the IDE catalytic chamber and its substrates. Thus, substrates cannot enter the cavity when IDE is closed, which is why IDE needs to undergo an opening state to capture its substrates inside the catalytic chamber. IDE open form is also required for the exit of proteolytic products. On the other hand, substrates can also lock IDE in the closed conformation to efficiently activate the proteolytic process [24]. 

Consistent with this, IDE is known to exist as an equilibrium of monomers, dimers, and oligomers [25]. Although IDE majorly exists as a homodimer and is more active than the monomeric form, the latter form does retain its enzymatic activity (Figure 1) [25]. Each monomer has 1019 amino acids and consists of four domains. An extended 26-residue loop connects the N-terminal domains (IDE-N: D1 and D2) and the C-terminal domains (IDE-C: D3 and D4). The N-terminal domains (IDE-N) D1 (residues 43–285) and D2 (residues 286–515) contain several charged, polar, and hydrophobic patches [24]. The surface of IDE-N is also largely neutral or negatively charged [24]. The catalytic site (residues H108, E111, H112, and E189) is located in IDE-N [24]. The exosite (residues 336–342 and 359–363), which is also present in IDE-N, is located approximately 30 Å from the crypt. This exosite is a key site in positioning peptides before degradation takes place. It is also the major site where the N-terminus of substrates are anchored [3,17,24]. On the other hand, the C-terminal domains (IDE-C) D3 (residues 542–768) and D4 (residues 769–1019) have a positively charged surface [24]. Although the catalytic site is situated in IDE-N, IDE-C is crucial for substrate recognition. Studies have shown that both IDE-N and IDE-C are essential for IDE activity and mutations of catalytic residues can severely decrease its function. For instance, IDE E111Q mutation renders the protein nonfunctional [26]. Moreover, site-directed mutagenesis of IDE H108 (H108L and H108Q) inhibits IDE catalytic activity but retains its ability to bind insulin [27]. Similarly, mutation of R824 and Y831 to alanine significantly reduces the catalytic rate of IDE [24]. 

IDE is a very challenging protein. Hence, the mechanism through which peptides are recognized is still elusive. The different forms that IDE structure adopts (open-closed) during the catalytic cycle need further investigation as well. Another challenging aspect is the detailed mechanism of IDE allostery that also remains unsolved [24,28]. Solving the detailed and defined allosteric path along the discovery of specific and potent allosteric modulators, triggering the activity of IDE, awaits future work. Moreover, only the closed and holo-conformations of IDE have been solved [19,20,21,22,23,24]. As for the open state, it has been assessed with Fab-assisted CryoEM at best (Appendix A) [28]. 

The current information about the conformational changes of IDE, as well as its dynamics information at the atomic level, is not sufficient yet. Moreover, although IDE makes the perfect therapeutic target for both type 2 diabetes and Alzheimer’s disease, its dual linkage can also be a challenge for the development of potent modulators. For instance, inhibition of IDE might raise a potential issue of an adverse effect, which would prevent its action to cleave misfolded and amyloidogenic peptides, such as amyloid beta. IDE modulators will also require a long-term evaluation to avoid the adverse effects [18]. Therefore, it is essential to further explore the atomic-level molecular mechanism involved in the structural transitions of IDE for the development of efficient, but also selective inhibitors, and uncover the substrate recognition process that might hold the key to cure many diseases [29,30]. This information would also be crucial to finally decipher the complete role of IDE. 

Here, we combine different bioinformatics approaches, such as structural modeling and molecular dynamics (MD) simulations, to address these questions and to complement the existing information concerning IDE and its structural mechanism. 

## 2. Results and Discussion

### 2.1. MD Simulations Analysis

IDE biological function is directly related to its conformational transitions. With that in mind, we ran 7 MD simulations of the IDE monomer in its unbound state to recover the protein structural changes. Hence, the total simulation time for this system is 7.5 μs. 

#### 2.1.1. Root Mean Square Deviation (RMSD) Evaluation of IDE Structure

To analyze the stability of each system, we performed a RMSD analysis of all our MD trajectories (Figure 2). In our study, seven systems were simulated for a total of at least 1 μs. However, run 4 was simulated for a longer time (1.5 μs) since it displayed extremely high RMSD values until 1 μs. As shown in Figure 2A, C*α* RMSD was found to stabilize the IDE system with values reaching 2.5 to 4 Å, with fewer fluctuations for most of the trajectories. On the other hand, the fourth and fifth trajectories displayed the largest fluctuations compared to the others. 

The RMSD values of the fourth run increased towards 600 ns until they reached their highest values at 1 μs then decreased at 1.1 μs to form a plateau until the end of the trajectory. Thus, the behavior of IDE in run 4 may have been due to the exploration of another state. This state is different from the initial closed structure of IDE, which might explain the excessive fluctuations in the RMSD values. Consequently, the large crypt movements and the flexibility of IDE can be explained through the exploration of an open state.

The RMSD values of the fifth run also increased, starting from 300 ns to reach a plateau with a constant value of 5.0 Å. This result also indicates a conformational change in the structure of IDE. With these observations in mind, for each of the third, fourth, and fifth runs, we extracted a frame at 1 μs, where IDE fluctuates the most. This relates to the results gathered from the RMSD with the changes occurring in the structure of IDE. As shown in Figure 2A, a noticeable twisting motion on D1 and D4 differentiated IDE of the fifth run from the other extracted frames. The helices and loops at the entry of the catalytic chamber witnessed a slight twist while D2 and D3 remained rigid. These movements can be described as rigid-body swing motions of the IDE gate [30]. It was fully explained by McCord et al. [30] that this twisting motion is characterized by a small rigid movement of D1 moving away from D4. Additionally, these movements were further analyzed with a quasi-harmonic analysis to confirm these observations (data not shown). In our case, this distinct state cannot be considered as open enough to enclose short peptides and did not differ significantly from the closed state, as only very small movements occurred in IDE. However, it is an interesting observation, and it enhances the fact that IDE can exist in a mixture of different conformations, and can explore several transient states, while remaining stable. 

Altogether, RMSD results showed that IDE exhibited interesting variations. These RMSD fluctuations can be correlated to the structural changes that IDE adopts to be stabilized. Moreover, RMSD fluctuations can be directly linked with IDE cavity changes, which are responsible for the protein structural rearrangements.

#### 2.1.2. IDE Cavity Volume and Hydration Analysis

Accordingly, the cavity volumes of IDE were calculated to examine the expansions of its structural flexibility and potential different states. As shown in Figure 2B, all trajectories, apart from the fourth, explored the same volumes with values ranging between ~15,000 and ~25,000 A^3^. The fourth run reached its highest value at 950 ns with a volume of ~35,000 A^3^, then a prompt decrease appeared at 1.2 μs, which ended in a plateau until the end of the simulation. Thus, these results are perfectly correlated with the RMSD values previously observed. Considering that the cavity of IDE has an initial volume of ~15,000 A^3^ in its closed state [15], the frequency distributions of IDE volumes were calculated and plotted in Figure 2C. Similarly, most of the trajectories displayed volumes corresponding to closed or semi-open states. On the other hand, the fourth run clearly showed the exploration of at least two different states. Indeed, the trajectory is divided into two separate and uneven populations. A more occupied population of volumes defined the closed or semi-open state, whereas a lesser population represented the open state. Therefore, IDE explored at least three different states (closed, semi-open, and open) during the MD simulations. We hypothesize that most of the trajectories have met the closed and semi-open states while the fourth trajectory, which displayed volumes more than twice the initial one, has met the fully open state. With the combination of these results, the IDE open-closed switch represented in the fourth trajectory was displayed (Appendix A) to capture IDE movements. 

Water molecules are important components in maintaining the functions of proteins. Since the RMSD and the cavity volume analyzes indicated major structural changes in the IDE structure, the solvent molecules, and the total solvent accessible surface area (SASA) were calculated for all trajectories. The SASA analysis stands for the solvent accessible area. Low values or a decrease in the SASA indicate a closed state of the protein structure with very few hydrophobic areas accessible to the solvent. On the other hand, high values, or an increase in the SASA, describe a certain degree of protein’s flexibility and the strong exposure of the cavity to the aqueous environment of the system. Thus, the higher the values of SASA, the more the cavity is exposed to the solvent and witnesses several conformational changes. As shown in Appendix A, SASA values mostly ranged between 39,000 Å^2^ and 48,000 Å^2^, with various fluctuations. As expected, the fourth system displayed the most important values of SASA with its highest value reaching ~49,000 Å^2^ corresponding to the expansion of IDE cavity. The fourth system also witnesses an important decrease of the SASA that correlates with the drop of the solvent molecules (Appendix A). This event was due to an IDE cavity volume decrease that was accompanied by the simultaneous expulsion of the water molecules. Thus, the SASA analysis, together with the RMSD and the volume cavity calculations, summarize that these results were coupled to protein conformational changes. These results also support the hypothesis that IDE switches from a closed to an open state through an allosteric behavior. 

#### 2.1.3. Root Mean Square Fluctuation (RMSF) of IDE Structure

To further analyze the flexibility and local changes in the structure of IDE, C*α* RMSF of each residue has been calculated (Figure 2D). RMSF analysis revealed that D1, D4, and an adjacent region to the linker are the most flexible parts of IDE. These fluctuations are mostly observed for the fourth run. C*α* RMSF values reached about 4 Å compared to the rest of the MD trajectories (2.0–2.5 Å). These observations agree with experimental data where the swinging door of IDE is mostly carried out by the movements of both D1 and D4, which are the principal regions causing the protein to undergo different states [28,30]. Appendix A illustrates the most flexible regions of the protein along with the residues exhibiting the highest C*α* RMSF values. Interestingly, the represented amino acids (Figure 2D) displayed the same pattern of fluctuations in all the trajectories, but with higher C*α* RMSF values for the fourth one. As expected, most of the residues are positioned on solvent-exposed regions, such as loops (Appendix A). Therefore, the high fluctuations of these residues can be explained through their intramolecular and intermolecular interactions within the protein and with the solvent. The most flexible residues positioned on D1 and D4 are also solvent exposed but were observed to play major roles in maintaining D1–D4 interactions along the IDE gate. For example, residue Q828 (D4) has been identified to be a key residue interacting with different residues of D1 (R181, E182, and N184) [24]. Residue Q828 exhibits a C*α* RMSF value of 4.1 Å when IDE is open against a value of 2.1 Å when IDE is closed. Accordingly, in the closed conformation of IDE, residues R181, E182, and N184 display values of 1 Å, 0.8 Å, and 1 Å, respectively. However, these residues present higher values when IDE is open with values reaching 4.3 Å, 3.8 Å, and 4.0 Å, respectively. Additionally, the flexibility of the exosite and the catalytic site residues were also examined. For all trajectories, both regions exhibited very low C*α* RMSF in all trajectories, which did not exceed 1 Å, indicating their structural stability. 

To support these results, the distance between the center of mass (COM) of D1 and D4 was plotted in Figure 2E. Interestingly, we observed the same pattern as the previous graphs. D1 and D4 were seen to be moving closer to each other at the beginning of the MD trajectory. Then, both domains moved away to reach their maximum distance value at ~59 Å followed by the recovery of their initial distance towards 1.1 μs. As shown in Figure 2F, the different major states of IDE are illustrated in the complementation of previous results. Furthermore, to rule out the hypothesis that the IDE open state might be an artifact, we analyzed the backbone RMSD of each domain during the fourth trajectory (Appendix A), which appeared to be stable along the simulation time. It is a simple way that serves as an indicator of conformational stability in the system during the simulation. 

#### 2.1.4. IDE Hinge Dynamics Analysis

IDE must undergo a hinge-like motion to transition from a closed to an open conformation. This transition is required for the entry of substrates and the release of proteolytic products. IDE possesses a hinge loop or a linker (516–541) connecting D2 and D3. This linker is critical for the proper pivoting motion between the IDE-N and IDE-C. Therefore, the hinge loop is an important region to regulate the allostery of IDE. 

To study the hinge loop dynamics and its role in the domains pivoting movements, we used a quasi-harmonic approximation implemented in the gmx anaeig module of GROMACS [31]. C*α* atoms of IDE were selected to carry out the analysis. We compared only the movements projected on the first eigenvector as they exposed the major differences. Figure 3A shows the superposed extreme projections of the linker along the first eigenvector. As shown in Figure 3A, we compared the behavior of the major motions of the hinge-loop, both when IDE is open and closed. Surprisingly, the loop remained stable and only slight movements can be noticed in the hinge. In the closed conformation, the hinge displayed minimal back and forth movements coinciding with the protein domain fluctuations. As for the open conformation, the linker represents more fluctuations as it moves along with the extension of the cavity. Accordingly, Figure 3B shows the C*α* RMSF evaluation of the linker in each run after extraction of the frames of the quasi-harmonic analysis. The loop exposed higher flexibility in the open conformation, compared to the closed one, with RMSF values ranging between 1.0 and 2.0 Å. The main explanation as to how the loop stays stable is because of the very tight interactions made within the protein. Indeed, the linker is a conserved region of the M16 metalloproteinase proteins [30,32]. 

Almost every residue of the loop interacts with either the D2 or the D3 domain through hydrogen bonding, hydrophobic interactions, and salt bridges (Appendix A). For example, K521, K527, and E541 form salt bridges with residues E349, E529, and K735, respectively. Several hydrogen bonds are formed among N528, E413, and A610. E536 and L538 form two hydrogen bonds with N732. T533 interacts with D636 and K637. Similarly, residue N543 interacts with D636 as well. As for E541, it also interacts with Q563 through hydrogen bonding. There are multiple hydrophobic patches formed with D2 and D3 through the hinge loop. Residue L524 forms hydrophobic interactions with L401 and W409. F530, which is a crucial residue for the maintenance of the catalytic role of IDE [30], interacts with Y607, A611, L616, I640, and L641. Finally, F535 interacts with V420 and F424. These extensive interactions are also conserved in both the closed and open state of IDE; hence, the preservation of a stable linker structure. 

Next, we measured the opening angle of IDE to describe, in further detail, the hinge-type motion (Figure 3C–E). Taking the COM of D1 and D4, combined with the COM of the linker, yields an opening angle of a maximum of 106 degrees (run 4) compared to ~68 degrees when it is closed (runs 1, 2, 3, 6, and 7). As expected, the angle values of the fourth trajectory follow the same pattern as the previous results (Figure 2A,F). The angle values fluctuate extensively until reaching their maximum at ~990 ns. A prompt decrease is observed at 1.1 μs, coinciding with the closing of the IDE cavity. Interestingly, the swinging door state (run 5) displayed a distinct angle spanning between 75 and 80 degrees. Therefore, the latter observation confirms the slight opening and twisting motions characterizing the swinging door motion of IDE. This leads to the conclusion that the description of the opening angle, using the hinge-loop as the center point, results in a more accurate distinction between the open and closed states.

#### 2.1.5. Gibbs Free Energy Landscape Analysis

Protein allostery is fundamental to understanding protein functions. Intra-protein atoms distances work via many allosteric processes with a defined path and the catalytic activity of IDE was proposed to be allosterically regulated in several papers [33,34,35]. However, the detailed mechanism of IDE allostery remains unknown. Here, we describe and retrace the allosteric communication of IDE as a series of local structural changes using the free energy landscape (FEL) approach. To study IDE dynamics movements, all trajectories were concatenated and the final free energy landscape for the first two most contributing principal components (PC) were calculated. To achieve this, a covariance matrix is constructed using the protein backbone coordinates. Then, the diagonalization of this matrix yields a set of eigenvectors and eigenvalues describing the collective modes of the fluctuations of the protein. Generally, the eigenvector with the largest eigenvalue or PC represents the large-amplitude collective motions of the protein. Since we have a system displaying significant movements, we selected the first two PCs characterizing these dominant motions. Moreover, it is very important that protein systems are locally equilibrated, and the determined pathways do not represent artifacts of the chosen coordinates. Thus, given the previous results, the FEL analysis was only used for the converged trajectories, which excluded the fourth MD simulation. These trajectories were concatenated into one single MD simulation to produce the FEL map. 

The FEL of IDE is shown in Figure 4A. The lowest energies are represented in blue, whereas high energies are indicated in red. This means that the blue regions represent stable states of the protein, while the red areas describe the unstable states explored during the MD simulations. Moreover, several bins with minimum energies mean that the protein explores different conformations through transition states. Accordingly, the global minimum conformations of IDE were extracted regarding the stable and unstable FEL bins to differentiate them. 

The projection of the backbone trajectories along the PCs revealed three major bins ((I), (II), and (III)). The investigation of these bins revealed that the first bin enclosed the structures of the first, third, second, and seventh trajectories, making it the major energy minima and the most occupied basin. Interestingly, the sixth system visited the first bin from 0 to ~47 ns, then jumped off from the major local minimum (I) and moved towards a more distant region of the conformational space, the second basin (II). Additionally, the fifth system explored the first basin (I) from 0 to 300 ns, then transited to the third basin (III) corresponding to the RMSD jump observed in Figure 2A. 

Visual examination of the lowest energy structures belonging to their corresponding bins revealed a very similar structure (Figure 4B). Thus, the closed conformations belonged to a single state but their mapping onto different space coordinates shows that structural changes appeared during the MD simulations. Therefore, to understand the conformational changes reflecting the different bins, we extracted one representative IDE structure from each bin (Figure 4B). Compared to most of the MD simulations, the sixth system revealed a slight greater flexibility in the alpha helices of the different domains, describing the constant dynamic movements of the protein. Overall, an RMSD of 2.0 Å was calculated between state (I) and state (II). For the fifth system, a higher RMSD value (2.6 Å) was calculated between state (I) and state (III). The fifth trajectory has already been described in the previous results as a different state on its own with a specific dynamic motion displaying a particular angle between D1 and D4. Therefore, the FEL map regrouped the different structural changes of IDE in terms of energy and highlighted important transitions in the cycle of IDE. 

#### 2.1.6. Non-Covalent Interactions

Residue interactions play major roles in IDE dynamic movements and allostery. Knowing that IDE is a flexible protein, it is very important to check the stability of the hydrogen bonds using MD simulations rather than inspecting only the crystal. Thus, the number of hydrogen bonds (HB) between D1 and D4, and between D2 and D3 were calculated (Figure 5 and Appendix A). Notably, the number of HB varied throughout the simulations and significant changes were observed. 

In most of the trajectories, the number of HB between D1 and D4 remained approximately stable and witnessed only little fluctuations. For example, for all the trajectories, the number of hydrogen bonds averaged between 5 and 7 (Figure 5). However, during the fourth trajectory, we can clearly observe a total disruption coinciding with the frame time of IDE open state exploration (900 ns–1.1 μs). Indeed, the drop of the number of HB, going from ~15 to 0, clearly defined the closed-open switch of IDE that was similarly witnessed in the previous results. Then, a recovery of the number of HB can be seen around 1.1 μs (8 HB) when IDE regains its closed conformation. The same analysis was applied for the D2–D3 complex (Appendix A). Similarly, D2 and D3 HB were stable during the MD simulations. The number of bonds averaged between 7 and 9. As expected, the fourth trajectory also displayed a decrease of the number of HB at the same frame time of D1–D4 HB disruption. However, this decrease was less important than the one observed between D1 and D4. Therefore, these results reinforce the fact that D1 and D4 are the main “gate” domains of the IDE closed-open switch. These results also suggest that D2 and D3 are still sustained during the open switch of IDE to maintain a certain stability of the protein. 

To identify the residues involved in HB formation between the different domains of IDE, HB occupancy was calculated. In Table 1, HB and salt-bridges (SB) observed across D1–D4 and D2–D3 binding interfaces are listed, together with their average occupancy percentage during the simulations. We isolated only the most frequent interactions with a threshold of 10% and with a cutoff distance of 4.0 Å.

Four pairs of residues (K898-D84, K884-E133, R824-E182, and K85-D895) formed SB between D1 and D4. The salt bridge formed by K898 and D84 remained stable for all the trajectories (Figure 6). In run 5, the HB occupancy was particularly high due to the angle formed by D1 and D4 (Figure 3). This angle, creating a favorable interaction together with an important lifetime K898–D84 bond, was completely disrupted during run 4. However, it recovered completely when IDE regained its closed conformation at the end of the MD simulation. Residue K884 formed a stable SB with E133. The SB occupancy ranged from 59.7% to 100.0%, attesting to the bond strength and sustainability. The bond created between R824 and E182 was positioned at the main gate of IDE. Interestingly, the R824–E182 bond was observed to be maintained in all trajectories, except for the fifth one. The swinging door motion did not allow the bond to be formed since D1 and D4 might have been too far away from each other and too flexible. As for the fourth trajectory, the R824-E182 interaction witnessed a low percentage of occupancy due to the IDE closed state recovery. The S132-E817 bond was positioned on two helices of IDE that were only accessible through the bottom side of the protein (Figure 6). The HB was well maintained in all simulations (34.1% to 82.3%) except for runs 4 and 5. As expected, in this case, the two domains were not close enough to form the HB. The fourth SB formed with K85 and D895 was also stable with higher occupancy when IDE was closed. Finally, Q828, which is a key residue positioned at the main gate of IDE, was revealed to interact with N184 and R181. Yet, Q828 was observed to form a stable bond only in a few simulations. This appears to be related to the high flexibility of the residue (Figure 1D and Appendix A). Altogether, the interaction patterns across the D1 and D4 interface agree with those known from experiments [24] and the main interactions were recovered in our analysis.

Compared to D1 and D4, more residues were observed to maintain the stability of the D2–D3 domains and 7 SB were formed in the D2–D3 complex (Appendix A). Residue D309 interacted with several residues, excepts in simulations 3 and 7. D309 was found to interact with N672 through its backbone in runs 2, 4, 5, and 6. The absence of this interaction was compensated by the interaction of D309 with either N671 or R668. K657 formed a stable SB with E382 in all the trajectories apart for the fifth run, where K657 interacted with E381 (96.7%) instead. Additionally, K571 connected the D2–D3 complex in all MD simulations with residues D426 and F424, where they occupied approximately 80% and 40% of the simulation time, respectively. It is also important to note that the SB contributed by K351 interplay with E606 and D602 to stabilize the protein. Moreover, the H336-Y609 stable HB interaction was mainly found in all simulations. Interestingly, H336 was replaced by H340 in the second trajectory to connect with Y609 (22.7%). Finally, residue R311 formed a stable SB with E664, the interaction occupied approximately ~70–100% of the trajectories. On the other hand, R311-E664 did not appear possible in the fifth simulation due to the angle formed by the swinging door conformation. As for the sixth run, R311 preferably interacted with R668 (21.7%) and, alternatively, E664 formed an HB with E381 (~37%) (data not shown). 

To further explore the dynamics and the movements of the residues relative to IDE allostery and support the previous results, we calculated the COM distance of each pair of residues along each trajectory (Appendix A). As shown in Appendix A, residues K888-E133, D895-K85, and K898-D84 displayed the most stable distances, which agree with the results observed in Table 1 and Figure 6. Apart from the fourth simulation (~30.0–48.0 Å), the COM distances exhibited little fluctuations and did not exceed 10.0 Å. Thus, these observations support the sustainability of K888–E133, D895–K85, and K898–D84 interactions within the D1 and D4 domains. As expected, the fourth simulation displayed a prompt increase representative of the opening of IDE, as tediously described in previous results. As for residues S132–E817, a clear increase of the COM distances can be observed for the fifth trajectory. Indeed, in the fifth MD simulation, residue, E817 does not interact with S132, hence the large distance between the two amino acids. Similarly, in this same trajectory, residue R824 and Q828 interact with E191 instead of E182 and N184, respectively. Thus, these observations explain the larger COM distance in run 5 for those residues.

The same analysis was applied on the D2–D3 complex in Appendix A. A first observation is that compared to the D1 and D4 interactions, little fluctuations are observed on the COM distance plots for D2 and D3. This result support the fact that D2 and D3 tend to follow rigid body movements among the large-scale motions of IDE. However, this does not exclude that a few residues witnessed higher fluctuations and larger COM distances. For example, D309-N672, D309-N672, D309-R668, E382-K657, E381-K657 and R311-E664 are the main residues displaying the largest distances (~20.0–25.0 Å). It should be noted that these amino acids are located on the front surface of IDE. On the other hand, residues D424-K571, F424-K571, K351-E606, K351-D602, H336-Y609 and H340-Y609, exhibited lower and stable COM distances (~10.0 Å). These residues are located on the back surface of IDE. Additionally, these same observations can be noticed for the D1–D4 complex. With these results, we suggest that fluctuations mostly occur in the front side of IDE while the back side and the rest of the protein is mainly stable. These results were interesting as they confirm that the main entrance of peptides and solvent is the most flexible region of the protein while the rest of the protein structure tend to have restricted motions to stabilize the ensemble of the four domains. Altogether, our results prove that the described residues are the main partaker in IDE dynamic movements and forming gate.

## 3. Materials and Methods

### 3.1. Protein Modeling

Homology modeling was performed to get the full-length structure and wild type/active form of the IDE protein [24]. For this purpose, the substrate-free 3D crystal structure of human IDE was retrieved from the RCSB Protein Data Bank in PDB format [36] (PDB ID: 2JG4, resolution: 2.80 Å) [37]. We selected the PDB ID: 2JG4 as it represents the only crystallized structure of the closed substrate-free IDE with the lowest resolution. Other IDE closed conformations can be found in the RCSB Protein Data Bank. However, these PDB structures are resolved with either Cryo-EM, FAB antibodies, peptides, or inhibitors, and can have poor resolutions. From this perspective, we fully focused our study on the closed form of the IDE free of substrates to concentrate on its conformational changes through MD simulations. 

The crystal structure contains one mutation at the catalytic site (Y831F). The structure also lacks residues 971–978 and 1012–1019 in the fourth domain. Therefore, the homology model building was carried out using the MODELLER software (v10.1) [38] with PDB:2JG4 as a template. Correct side-chain orientations of the catalytic site residues (H108, E111, H112, and E189) were verified to avoid any clash with other residues and the zinc atom. For validation, the model with the lowest value of the DOPE assessment score [39] was selected for further analysis and VERIFY 3D [40] was used for further endorsement (Appendix A). The final model comprised 15,880 atoms corresponding to the 977 residues (43–1019) of the full-length IDE. The protonation states of ionizable residues in the IDE model were assigned using PROPKA [41] and the pH was set at a physiological value of 7.5. 

### 3.2. Molecular Dynamics Simulations 

We ran 7 large-scale simulations of 1 to 1.5 μs each. MD simulations were carried out with the GROMACS (v2019.5) software package [31] and we used the CHARMM36m force field [42] for both the protein and zinc parameters. The protein was placed in a dodecahedron-shaped water box (TIP3P) and a minimum of 10 Å was preserved between each atom of the system and the walls of the box. All runs were run at a 300 K temperature and with a time step of 2 fs. All bonds were constrained using the LINCS algorithm [43] for the protein and the SETTLE algorithm [44] for the water. The energy of the system was minimized over 1000 steps using the Steepest descent algorithm after ion addition and system neutralization. The minimization convergence was set at a maximum force of 1000 kJ/moL/nm. The number of particles, volume, and temperature (NVT) equilibration was performed for 1000 ps at a temperature of 300 K with a coupling constant of 0.1 ps. number of particles, pressure, and temperature (NPT) was run by setting the temperature to 300 K and the pressure to 1 bar. Electrostatic forces were calculated with the particle-mesh Ewald algorithm [45]. For trajectory analysis, we used GROMACS packages, VMD (v. 1. 9. 4a38) [46], PyMOL [47], and GNUPLOT (v. 5. 2) [48]. The volume of the IDE crypt, as well as the number of solvent molecules, were computed using the trj_cavity_v2.0 program [49]. 

RMSD and RMSF analyses were led with GROMACS. The RMSD analysis measures average distances (in Å) of the studied systems from the corresponding starting structure over the simulation period. RMSD is defined as:$$\mathrm{RMSD}\left(t\right)={\left[\frac{1}{M}\Sigma_{i=1}^{N}m_{i}|r_{i}\left(t\right)-r_{i}^{\mathrm{ref}}{|}^{2}\right]}^{1/2}$$
where $M=\Sigma_{i}m_{i\;}$ is the total mass, $m_{i}$ is the mass of atom i, N is the number of atoms, and $r_{i}\left(t\right)$ is the position of atom i at time t after least square fitting the structure to the reference structure. In our study, the calculation and fitting of the protein were done on the C*α* atoms. 

To examine the flexibility and the local changes in the structure, C*α* RMSF versus the number of IDE systems were investigated. The RMSF equation is defined as:$${\mathrm{RMSF}}_{i}={\left[\frac{1}{T}\Sigma_{t_{j}=1}^{N}|r_{i}\left(t_{j}\right)-r_{i}^{\mathrm{ref}}{|}^{2}\right]}^{1/2}$$
where T is the time over which the mean coordinate is calculated regarding the $r_{i}^{\mathrm{ref}}$ (reference position of particle i). The RMSF was computed from the atomic coordinates of the C*α* atoms as well. 

### 3.3. Gibbs Free Energy Landscapes 

The Gibbs Free energy landscapes were performed to describe IDE motions and different states through the simulations. We applied the gmx covar, gmx anaeig, and gmx sham modules of GROMACS to calculate the two-dimensional representations of the FEL. The FEL of each run was constructed using the projections of their first (PC1) and second principal components (PC2) or eigenvectors.
$$\Delta G\left(PC1,PC2\right)=-{Κ}_{B}Tln\;P\left(PC1,PC2\right)$$
where ${Κ}_{B}$ is the Boltzmann constant, T is the temperature of the simulation, P(PC1, PC2) illustrates the probability distribution of the system along with the first two principal components. PC1 and PC2 usually display the dominant fluctuations in residues for a protein, hence why they were studied in this project. 

## 4. Conclusions

The catalytic activity of IDE is mediated by several structural transitions. These conformational changes are allosterically regulated by its substrates, ATP, and other protein partners [10,33,34,35,50]. In this paper, we suggest that IDE allostery is conducted by the collective motions of the numerous atoms forming IDE structure. Based on our data, we present a model of IDE in its active state to explain how the allostery of IDE is coordinated by the equilibrium between IDE closed and open states. Our MD simulations revealed that IDE undergoes several states and is a flexible system. RMSD (Figure 2A) and IDE cavity volume calculations have put forth at least three different explored conformations. These different conformations are defined by a closed, a swinging door, semi-open (intermediate state), and an open state. We believe that the probability to recurrently exploring IDE states or new conformational changes can increase and enhance with further computational methods. Notably, future work can focus on the required transition energy to switch from a closed state to an open state with additional MD sampling.

Interestingly, all states were explored during the fourth trajectory with a complete cycle of IDE closed–open–closed transition. Such motions can explain how allostery regulates and governs the IDE biological structure but also suggest additional ways in how IDE may function. Furthermore, the swinging door motion state [30] was explored by one of the simulated trajectories (RMSD: ~4.0–5.0 Å (Figure 2A)). Thus, we completed these results with an RMSF analysis to define the dynamic domains responsible for IDE flexible movements (Figure 2D). RMSF results revealed that D1 (RMSF: ~4.5 Å) and D4 (RMSF: ~4.0 Å) are the main actors of IDE distinct motions, which agree with different published studies [26,30]. These results were supported by the calculation of IDE cavity volumes over time (Figure 2B). The description of the volumes described the different states explored during the MD simulations. The closed and semi-open state volumes ranged between ~15,000 and ~25,000 A^3^, while the open state reached higher values (~15,000 A^3^ to ~35,000 A^3^). Thus, our data agree with the experimental data, where the average cavity volumes (~35,600 A^3^) and domain distances reaching their highest values at ~37 Å and ~55 Å are very similar to the simulated semi-open and open states, respectively [26]. Additionally, the cavity analysis was complemented with a SASA study (Appendix A). Moreover, we investigated the hinge dynamics through a quasi-harmonic study (Figure 3). The linker connecting IDE-N and IDE-C was shown to be extensively flexible as the IDE cavity opens, yet stable enough to maintain D2 and D3 connected. Additionally, the angles formed between the D1-linker-D4 in the different simulations were calculated with the aim of supporting the previous results (Figure 3). The determined angles were 68°, 87°, and 106° for the closed, swinging door, and the open sate, respectively. The obtained results confirmed that the angles match with the different structural states observed during MD. It was also observed that IDE is a very dynamic protein and can go through several conformational changes. Therefore, we ran a Gibbs FEL analysis to explore the stability of the explored states of IDE (Figure 4). Only converged MD simulations were used to construct the FEL map. Previous conformation changes observed in IDE protein were confirmed and explored in terms of energy in the FEL analysis. Three major bins were visited by IDE structures, displaying the protein dynamic and its closed state. Moreover, the swinging door motion of IDE was explored as a separate state, which confirms previous results. The open state is still a very challenging conformation to capture. Hence, these observations support the fact that IDE is unstable and needs a substrate to be stabilized in its open conformation. It is also known that extensive contacts are shared between IDE-N and IDE-C, so the IDE structure can be maintained (Figure 5). Thus, the residues maintaining these contacts, as well as the effect of IDE opening, were determined (Table 1 and Appendix A). With these results, several mutations, MD simulations, and docking studies with different IDE substrates can be directed to complete the full biological cycle and allostery mechanism of IDE. 

## Figures and Tables

**Figure 1 ijms-23-01746-f001:**
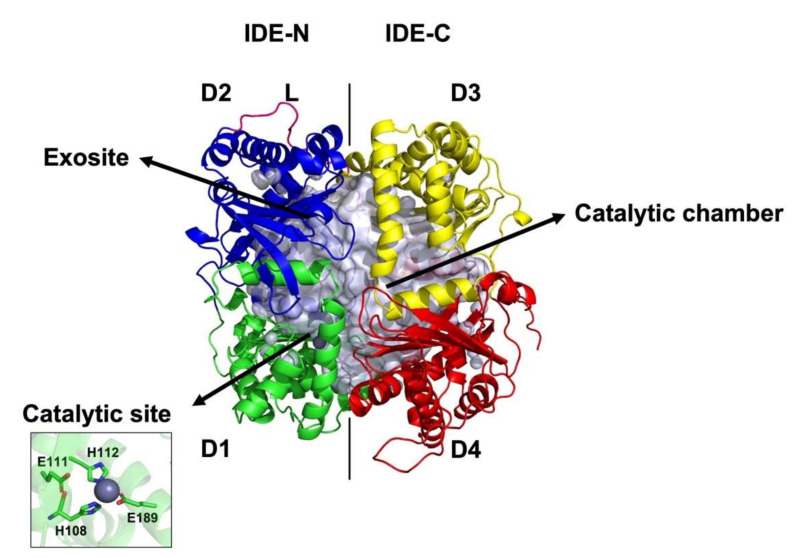
Representation of IDE structure and D1, D2, D3, and D4. IDE-N and IDE-C, as well as the linker (L), the exosite, and the catalytic site, are specified.

**Figure 2 ijms-23-01746-f002:**
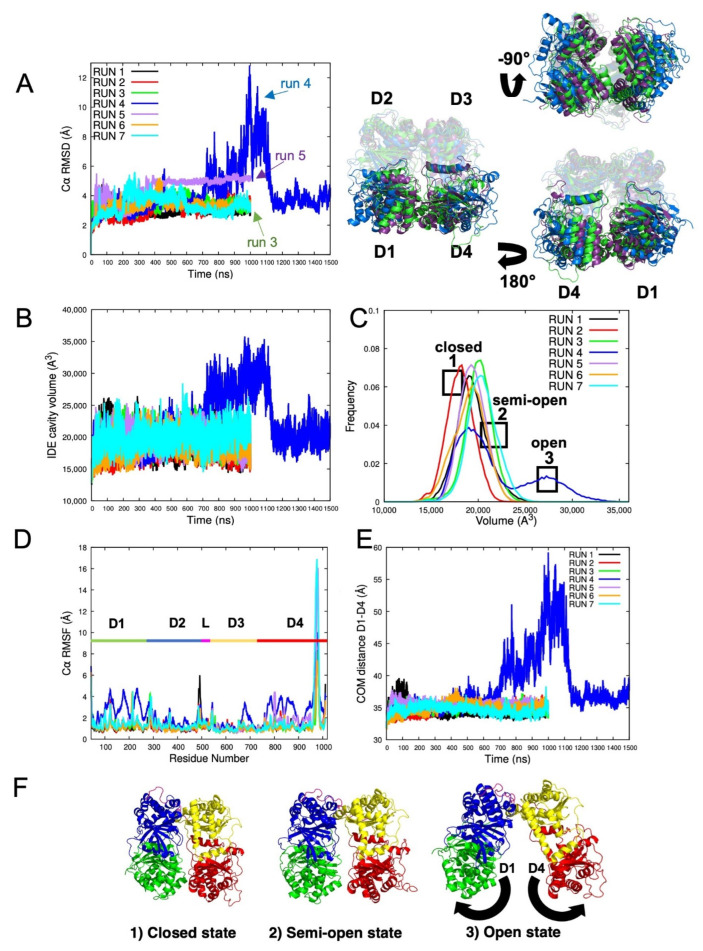
Evaluation of IDE MD simulations and major states of IDE. (**A**) Left: C*α* RMSD evaluation of IDE trajectories and comparison of IDE structures from run 3 (closed), run 4 (open), and run 5 (swinging door) at 1 μs. Right: domain structural analysis: only D1 and D4 are highlighted according to their designed run color, to see the conformational changes of the IDE door. (**B**) Evolution of IDE cavity volumes. (**C**) IDE cavity volume frequency for each MD trajectory. (**D**) C*α* RMSF of IDE different domains. (**E**) Center of mass distance between two domains: D1 and D4. (**F**) Major states of IDE explored during MD simulations.

**Figure 3 ijms-23-01746-f003:**
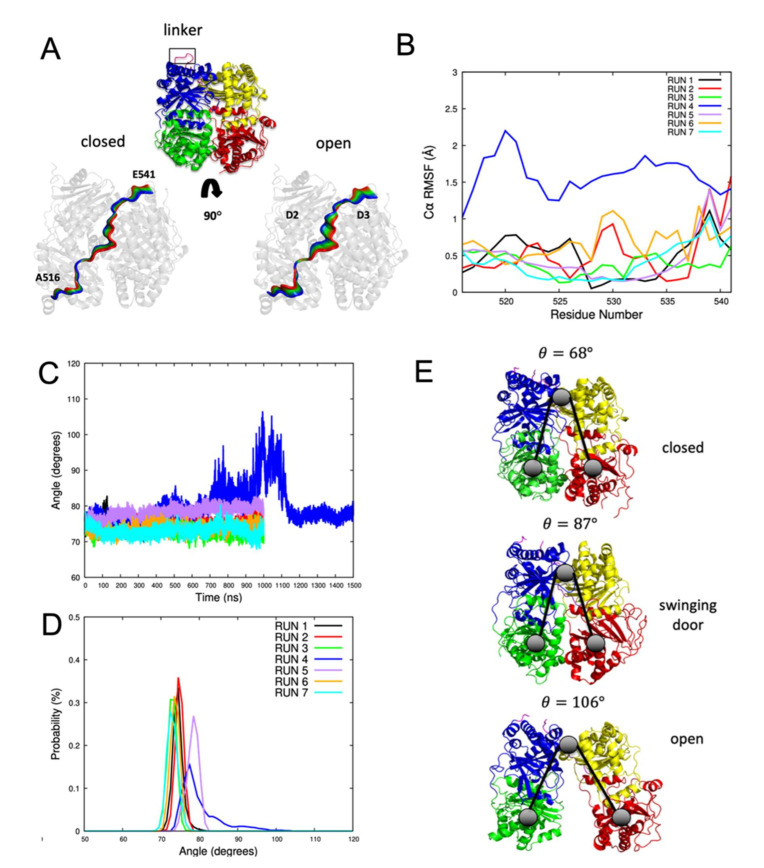
Quasi-harmonic analysis of the IDE hinge-loop. (**A**) The superposed frames of the third (closed) and fourth (open) trajectories are projected onto the first eigenvector. A total of 100 frames were sampled for each trajectory. Colors range from blue (first frame) to red (last frame). (**B**) Cα RMSF evaluation of the linker represented in the 100 sampled frames after the quasi-harmonic analysis. (**C**) Characterizing the hinge-loop motion and angle measurements by taking the COM of D1 and D4 combined with the COM of the linker. The angles were measured along the trajectories. (**D**) Probability of the angle values for each trajectory. (**E**) Representation of the most probable closing/opening angles.

**Figure 4 ijms-23-01746-f004:**
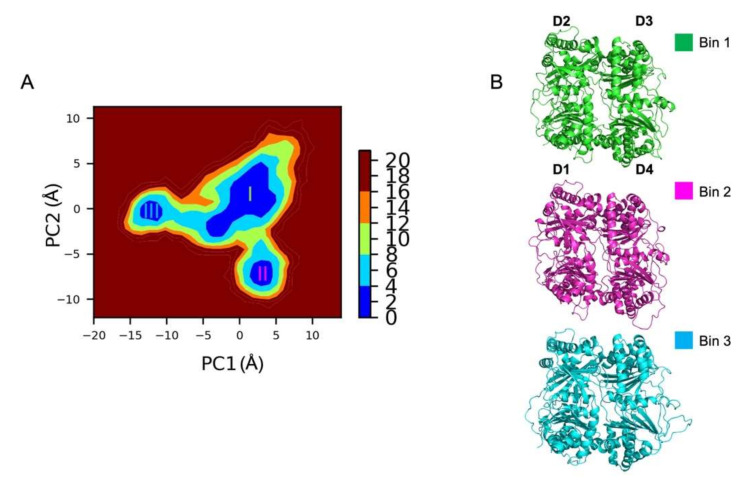
(**A**) Gibbs free energy landscape (FEL) analysis (**B**) with representative IDE structures extracted from the MD simulations. Free energy values are represented in kJ/mol, and their colors are detailed in the color bar. The representative IDE structures are represented in cartoons and display each basin.

**Figure 5 ijms-23-01746-f005:**
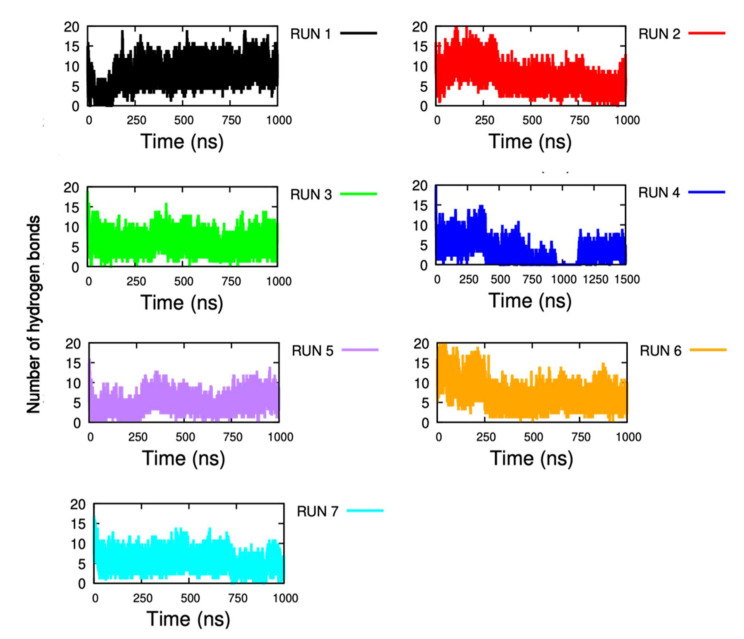
Number of hydrogen bonds formed between D1 and D4 along the MD simulations. Black, red, green, blue, purple, orange, and cyan are respectively for runs 1, 2, 3, 4, 5, 6, and 7.

**Figure 6 ijms-23-01746-f006:**
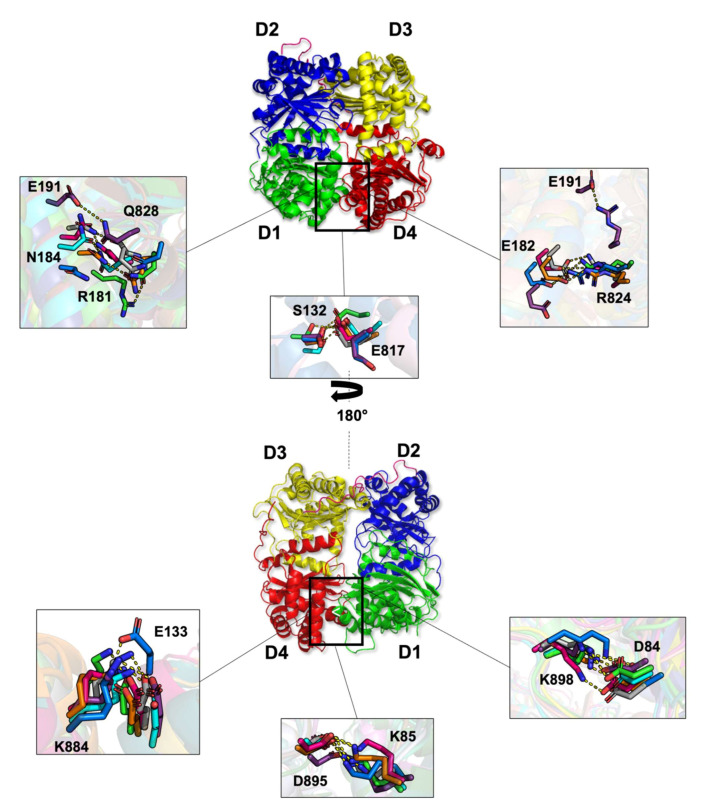
Hydrogen bonds and salt bridges formed between D1 and D4. The color code was conserved, according to the simulation number. Grey, red, green, blue, purple, orange, and cyan are respectively for runs 1, 2, 3, 4, 5, 6, and 7.

**Table 1 ijms-23-01746-t001:** D1–D4 non-covalent interactions occupancy (%) during MD simulations. HB and SB were reported only if they exist for >= 10% of the investigated period. Backbone (bb) and side chain (sd) interactions were specified.

Domain 1	Domain 4	Non-Covalent Interaction Type	Occupancy (%)						
			Run 1	Run 2	Run 3	Run 4	Run 5	Run 6	Run 7
D84 (sd)	K898 (sd)	SB	31.7	73.2	39.7	35.8	92.0	76.3	37.4
E133 (sd)	K884 (sd)	SB	100.0	98.0	100.0	73.1	59.7	100.0	100.0
E182 (sd)	R824 (sd)	SB	57.6	100.0	86.5	26.9	0.0	26.9	88.5
S132 (sd)	E817 (sd)	HB	82.3	39.4	34.1	0.0	0.0	35.8	27.1
K85 (sd)	D895 (sd)	SB	90.8	80.0	60.3	43.3	35.1	52.0	100.0
N184 (sd)	Q828 (bb)	HB	70.6	0.0	0.0	0.0	0.0	0.0	0.0
N184 (sd)	Q828 (sd)	HB	NA	0.0	0.0	0.0	0.0	11.0	0.0
R181 (sd)	Q828 (sd)	HB	NA	0.0	12.9	0.0	0.0	0.0	0.0
N184 (sd)	Q828 (sd)	HB	NA	14.9	0.0	0.0	0.0	0.0	0.0

## Data Availability

Data are contained within the article.

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
