# Peer review of "Exploring the Structural Rearrangements of the Human Insulin-Degrading Enzyme through Molecular Dynamics Simulations"

_ijms, 2022, doi:10.3390/ijms23031746_

Round 1
Reviewer 1 Report
The manuscript ‘Exploring the structural rearrangements of the human Insulin-Degrading Enzyme through Molecular Dynamics simulations’ by M. Ghoula et al. presents a thorough and precise analysis of IDE allosteric conformational changes from a close state to an open state using molecular dynamics simulation. The mechanism of opening suggested by the authors based on the complete analysis of 7 different simulations seems consistent with what was previously experimentally known. The authors have analyzed the behavior of several different parameters for the seven run in a very clever way. I particularly appreciate the Gibbs free energy analysis, which gives a clear view of the three different minima in the conformational landscape of IDE (Fig. 4A). Overall, this study gives new insight in the allosteric mechanism of this important enzyme.
However, the following questions should be addressed prior to publication.
- Two of the 7 simulations give insight to two different conformations different from the initial close one. It is not evident to understand if these two interesting simulations happen only by chance. What is the probabilities that a given simulation explores a new conformational state? Can we conclude that the open or the semi-open conformation has only 2 chances on 7 to occur without a substrate? And how a substrate would favor the opening of the cavity?
- There is a lack of information’s in the Introduction section that lead to confusion. From what is written and clearly presented, IDE prevents type II diabetes and moreover prevents toxic amyloid formation. The authors state at the end of the Introduction that their work is essential for the development of efficient inhibitors of IDE!!! Why do they want to inhibit an enzyme that could be efficient against diabetes and Alzheimer disease??? Please clarify.
- I would also like to have more information about the dimer formation and its implication on its function. It is written that IDE exist as a homodimer. Does IDE act only in the dimeric form? Is there an allosteric mechanism between the two monomers? What is the stoichiometry of the catalytic reaction? Amongst the most flexible residues, Lys 123 seems to belong to the dimeric interface (from the PDB file 2JG4). Does this interfacial position have an importance in the conformational modifications?
- The exosite is mentioned in the Introduction section and on the figure 1, but without any more information’s. Which residues are involved? Are they stable during the different simulations? Same questions for the catalytic site? The four residues which are bound to the zinc atom are stable whatever the simulations?
- The initial starting state for the different simulation is the PDB file 2JG4 which correspond to a close state. In the Introduction section, amongst the 6 references about the structures of IDE solved in the close state, the reference of 2JG4 is missing… Please add it. This crystal structure contains one mutation Y381F rendering IDE inactive. It would be nice to know how this mutation inactives IDE and what is the behavior of this residue during the simulation.
- There are many PDB structures of IDE in the PDB. The authors should explain their choice of the PDB file 2JG4. It would be nice to have a short comparison between the different PDB structures in the close conformation, which would explain the choice of 2JG4.
- I think there is a mistake in the sentence ‘as for the open state, it has been assessed with Fab-assisted Cryo-EM at best’ (line 63). The Fab-assisted structure of IDE in the open state is a crystallographic one, PDB 4IOF, ref [27] in the manuscript, while the cryo-EM structure f IDE in the open state is in complex with insulin, PDB files 6BF6 and 6BF8, ref [25] in the manuscript.
- It would be nice to have in the Conclusion section a more detailed comparison between the open state explored in the simulation and the open states described in different structures of IDE. It is only written: ‘which agree with different published studies [25, 27]’ (line 490). It would be nice to have more information about the large opening explored in the simulation and the experimental studies. Does the distance between the two domains are similar in the experimental structures and in the simulation? What is the size of the cavity in the experimental structures?
- Paragraph 2.1.2. IDE cavity volume and hydration analysis. I agree with the proposed analyses but less with the conclusion of this paragraph. The authors describe a decrease of SASA, a drop of solvent molecules, and a collapse of the IDE cavity which would occur after the open state. The different figures only show a return to the close state, I don’t see any collapse of the cavity. I don’t understand also why the drop of solvent molecules would be accompanied with a loss of hydrophobic interactions among non-polar residues connecting the different domains (line 148)
- Figure 2C: The signification of the square found on this figure is lacking. The open state of clearly evidence during the run 4. However, the semi-open state seems to occur, with respect to the cavity volume frequency, during the runs 3 and 7, and not during the run 5. Please explain.
- The exploration of the semi-open state during the run 4 is clearly shown with the analysis of the Ca rmsd (Fig. 2A). On Fig. 2D and Fig. S5, it seems that it is only the D4 domain which is flexible and moves away from D1. Is it an artefact of the Ca rmsf calculation?
Minor modifications
Figure 1 The legend of the figure 1 is wrong. It is the representation of a IDE structure with domains D1, D2, D4 and D4. The term ‘subunits’ is inappropriate. I also don’t understand why there is a vertical line in the figure? What is the meaning of this line?
Line 145 49,000 instead of 4,900
Line 281 Reference [44-46] : number not in the order on the manuscript. It seems that these three references have been added too late.
Line 264 1st occurrence of FEL abbreviation without explanation, should be written FEL (free energy landscape).
Table 1 I do not understand why the amino acid 1 and 2 do not always belong to the same domain. I would have written the Table 1 with the 1st column corresponding to the amino acids from D1 and the second column with the D4 amino acids. Same remarks for Table S1.
Figure S4D N841 instead of N481

Reviewer 2 Report
Recommendation: Publish after major revisions.
In this manuscript the authors studied Insulin-Degrading Enzyme (IDE) which function is rapidly brook down insulin and other peptides to prevent toxic amyloid formation . Thus, IDE plays a major role in preventing type II diabetes and other diseases like Alzheimer’s. The peptides are degraded by IDE in a large cavity (~15,000 A3), named crypt. The work presented in this manuscript aimed to study IDE structural changes and elucidate how IDE conformational dynamics can modulate the IDE catalytic cycle.
The manuscript should not be published in its present form. The authors must before clarify and corrected the following points.
Introduction:
An important feature of IDE is its large cavity (~15,000 A3) where peptides are degraded based on their size, charge distribution, and amyloidogenic nature [19-24]. Is it possible to indicate this cavity on figure 1?
Is it possible to present in figure 1 the known crystallographic, Cryo-EM conformations of IDE (closed, semi-open and open state)?
The following sentence should be rephrased: “IDE monomer consists of four domains connected through a 26 residues loop (residues 516-541) [24]”. The loop connects the block 1 (domain 1 and domain 2 – IDE-N) and block 2 (domain 3 and domain 4 – IDE-C) …
Studies have shown that both IDE-N and IDE-C are essential for IDE catalytic activity and mutations can severely decrease its function [24]. What type of mutation and in which domain?
The following sentence should be corrected: “The different forms that (?) IDE structure adopts (open-closed) during the catalytic cycle need further investigation as well.”
Another challenging aspect is the detailed mechanism of IDE allostery that also remains unsolved [24,25]. Is it possible to give more details of this allostery here?
RESULTS AND DISCUSSION:
As shown in the S2A figure, C? RMSD has been found to stabilize the IDE system with values reaching 2.5 to 4 Å with fewer fluctuations for most of the trajectories. The number of figure is not correct?
The SASA analysis was carried out on the whole protein or only on the cavity?
The number in following phrase is wrong 4,900 -> 49,000 Å2 (?) “As expected, the fourth system dis- played the most important values of SASA with its highest value reaching ~4,900 Å2 corresponding to the expansion of IDE cavity.
Figure 2A - It is impossible to visualize the movement of the domains in figure 2A. The figure must be changed. A presentation of the structures separately could improve the reading of this figure maybe?
The following sentence should be corrected: These trajectories have been concatenated into one single MD e simulation to produce the FEL map.
Figure 4B - It is impossible to visualize the movement of the domains in figure 4B. The figure must be changed.
The chapter on salt bridges and hydrogen bonds is too dense and difficult to follow. it will be better to rewrite this part to better highlight crucial information coming out from this part of analysis.
CONCLUSION
The following sentence should be corrected: The closed and semi-open state volumes ranged between ~15,000 and ~25,000 A3, while the open state reached higher values (~15,000 A3). ’15,000-> 35,000 A3)
Reviewer 3 Report
The manuscript by Ghoula et al reports on conformational dynamic study on IDE. It is certainly carried out in a fairly correct way, even though authors claim in the Abstract and in the Introduction to deal with IDE allostery without referring to allosteric modulators. Therefore, either they remove the term “allostery” (mentioning only “IDE conformational dynamics”), or else they make an effort to implement the Introduction with additional elements referrable to allosteric modulators.
From the technical viewpoint, in the Methods all parameters employed for the Zn binding site are missing; therefore, they must be clearly reported.
Minor points:
- At lane 145: I think that 4,900 Å is wrong, and probably 49,000 Å is correct.
- At lane 181: the sentence “Then both domains moved away by at least 25.0 Å” does not seem to correspond to what reported in Fig. 2E, since in the trajectory 4 the maximal excursion looks of about 50 Å.
- From lane 183 and 185 an RMSD analysis is cited but its routcome is not described and/or commented.
- In Figure 4B IDE domains should be labelled.
Round 2
Reviewer 3 Report
Accepted, since the authors answered satisfactorily to my questions and modified the manuscript accordingly